# Citizen Perceptions of Fake News in Spain: Socioeconomic, Demographic, and Ideological Differences

**David Blanco-Herrero** *[ID], **Javier J. Amores** [ID] and **Patricia Sánchez-Holgado** [ID]

Facultad de Ciencias Sociales, Campus Unamuno, University of Salamanca, 37007 Salamanca, Spain;
javieramores@usal.es (J.J.A.); patriciasanc@usal.es (P.S.-H.)
* Correspondence: david.blanco.herrero@usal.es; Tel.: +34-659-62-33-93

**Abstract:** Although the phenomenon of disinformation and, specifically, fake news has become especially serious and problematic, this phenomenon has not been widely addressed in academia from the perspective of consumers, who play a relevant role in the spread of this content. For that reason, the present study focuses on determining how this phenomenon is perceived by citizens, as the strategies to counteract fake news are affected by such opinions. Thus, the main objective of this study was to identify in which media the perception and experience of fake news is greatest and thus determine what platforms should be focused on to counteract this phenomenon. A survey was conducted in October 2020, among the Spanish adult population and was completed by a total of 423 people (with 421 valid answers). Among its main findings, this study determined that social media platforms are the type of media in which the greatest amount of fake news is perceived, which confirms the suggestions of previous studies. Furthermore, the experienced presence of fake news seems to be primarily affected by age and gender, as there was a higher level of skepticism observed among young people and women. Additionally, the use of media seems to be positively correlated with the perceived and experienced presence of fake news.

**Keywords:** fake news; disinformation; misinformation; hoaxes; social media; citizen perceptions

## 1. Introduction

According to the Trust in News study [1], 46% of news audiences believe that fake news influenced the outcome of recent elections. A Eurobarometer in 2018 [2] showed that 83% of respondents perceived fake news as a danger to democracy, while 53% of Spaniards claimed to encounter fake news daily or almost daily. An Ipsos study [3] further showed that 57% of Spaniards admitted to believing fake news.

Despite the relevance and interest of these observations, disinformation has not been commonly studied in academia from the perspective of citizens and consumers. The few authors who have studied this topic include Tandoc, Lim, and Ling [4]. These authors considered the citizen's role to be key, as citizens are the consumers of fake news and hoaxes. Indeed, understanding the opinions and experiences of citizens is essential to understand disinformation and present effective solutions. Consequently, the aim of the present study was to fill existing knowledge gaps on the perceptions of citizens regarding this problem. More specifically, this study did not analyze the factors influencing the diffusion or credibility of fake content, as this subject has been explored in the past [5–7]. Instead, we sought to discover how this phenomenon is perceived by citizens, given that media literacy and other strategies to counter fake news are affected by these opinions. Due to these potential practical applications, the main objective of this work was to discover in what types of media and social platforms the perception and experience of fake news is greatest, as these media require the strongest efforts to counteract disinformation. To enable a deeper analysis, a further aim of this study was to determine the existence of potential differences due to age, gender, level of education, socioeconomic level, and ideology.

Besides the interest of the topic of study, this article enriches existing quantitative literature by using a deep statistical approach to fake news in a Spanish setting. However, despite not being the most common approach, several works in Spanish academia have applied surveys related to our topic of study, such as the surveys developed by Gualda and Rúas and Masip, Suau, and Ruiz-Caballero [8–11]. Many works have also focused on more specific aspects, including several studies on fact-checking activities [12,13] and media analyses [14,15]. Additionally, many recent works focused on the COVID-19 pandemic [16–18], following an international trend. Building upon these works, the aim of the present study was to increase knowledge of fake news in a Spanish context.

This article addresses and narrows the discussion about fake news, especially in a Spanish setting. First, we detail the process followed to study the perceptions of Spanish citizens on this issue. Then, we explore the obtained results and, finally, discuss the results in connection to the existing literature in the field while also describing the study's limitations and future lines of work.

## 2. Literature Review

The first challenge in studying disinformation and fake news is selecting and defining the appropriate terms for this phenomenon. The largest discussion surrounds the use of the term "fake news". Authors such as Wardle and Derakhshan [19], in one of the most influential texts on disinformation, rejected the use of "fake news"—first, because this term is inadequate to describe the complex phenomenon of "information pollution", and second, because politicians from around the world have appropriated the term to describe news organizations whose coverage they dislike. Despite this rejection of the term "fake news", other authors have supported the use of this term in contrast to "disinformation"; such authors argue that both are valid but describe different realities. Bennett and Livingston [20] differentiated between "fake news" (isolated incidents of falsehood and confusion) and disinformation (more systematic disruptions of authoritative information flows due to strategic deceptions). Similarly, in a Spanish setting, Tuñón Navarro, Oleart, and Bouza García [21] highlighted the differences between traditional disinformation—the spread of information that is hard to verify and its subsequent use to obtain a variety of benefits—and the more novel concept of fake news—completely or partially fake information designed to look like real news, with the goal of confusing the audience and obtaining a political or economic profit. These authors also noted that fake news is a type of disinformation that has reached a high level of popularity, both in academia and public opinion.

Following this line of thought, the present work supports the use of both terms, as long as each term refers to its respective reality. Thus, our analysis focused on isolated cases where fake content attempts to imitate the format and style of journalistic news, rather than more complex disinformation campaigns. For this reason, we primarily use the term "fake news" (the equivalent Spanish term, *noticias falsas*, was used in the questionnaire).

Beyond a terminological discussion, it is relevant to highlight the growth of the problem of disinformation in recent years. This problem is not new, and much discussion around this phenomenon has focused on this factor [22,23]. Indeed, some of the explanations behind the proliferation of fake news utilize traditional theories, such as confirmation bias [24,25] and selective exposure [26], as well as traditional communication and media theories, such as agenda-setting [27]. However, other more novel elements and theories should also be considered, including the roles of bots [28] and, very importantly, the roles of echo chambers and filter bubbles [19,29]. These factors are strongly related to confirmation bias and selective exposure and have been widely discussed in relation to the social media environment [30].

Thus, despite its long-term existence, the current importance of fake news is unquestionable. Indeed, fake news is not only discussed by academia and professional media but also remains present in all aspects of communication and society [31]. The Spanish journalists surveyed by Blanco-Herrero and Arcila-Calderón [9] noted the seriousness of disinformation in the profession, and several other authors have shown that

this phenomenon has become one of the largest threats to democracy and society as a whole [11,20,32]. Bakir and McStay [33] also added that the situation generated by fake news is socially and democratically problematic: It leads to ill-informed citizens who are prone to remaining ill-informed in their echo chambers and becoming emotionally polarized and enraged due to the affective and provocative nature of fake news.

It should be also noted that, although fake news is usually associated with textual content, such news is also spread through different formats, including images and videos. The emerging technology of deepfakes [34] is one of the most relevant challenges in the current scenario [35]. However, there are already multiple efforts to combat fake news—such as through legislation and media literacy. Among these efforts, the relevance of automatic detection is garnering significant attention due to its already promising results [36]. Alternatives include crowdsourcing detection [37] and the well-established task of fact-checking [38].

Regarding the reasons for the increased importance of this problem, Figueira and Santos [23] claimed that to understand the particularities of present-day disinformation, two structural factors should be explored: the trust crisis in the media [7,39] and the appearance of a new and more complex media ecosystem. One of the most characteristic features of this ecosystem is *infoxication*, which is associated with the prevalence of infotainment, the exploitation of highly attractive topics, a lack of attention to journalistic ethics, and the pursuit of *viralization* [40]. This factor cannot be understood without examining the precariousness of journalism, which other studies [41,42] have already connected to lower ethical and quality standards, thereby contributing to a lack of trust in the media. Bakir and McStay [33], in one of the most complete approaches, explained the current phenomenon of fake news in connection with five factors derived from the ecology of digital media: the financial decay of traditional media; the immediacy of the digital environment; the creation and rapid circulation of misinformation and disinformation created with illegitimate goals or because of ignorance; the growing "emotionalization" of discourse; and the profit generated by the algorithms used in social media and search engines.

Besides these factors, the most relevant feature of fake news today is its broad circulation online. Indeed, during the U.S. Presidential Elections in 2016, fake news survived thanks to traffic on social media, which generated 41.8% of the visits to such news sites, in contrast to only 10.1% among the reference news sites [43]. There are, moreover, no precedents to the spreading capacity of information disorders on social media [26]. Vosoughi, Roy, and Aral [44], in one of the most relevant works in the field, observed that fake news has a 70% greater chance to be reproduced, as well as a further, faster, deeper, and broader reach than true news in all categories of information, mostly politics. Similar observations were made by Mathew et al. [45] and by Allcott, Gentzkow, and Yu [32].

Thus, even though the phenomenon of fake news affects the whole media system, social media seems to play a key role. This allowed us to formulate the following hypothesis:

**Hypothesis 1 (H1).** *Spanish citizens consider social media to have a greater presence of fake news than other media formats.*

More specifically, Allcott, Gentzkow, and Yu [32] and Silverman [46] empirically observed that during the U.S. Presidential Elections in 2016, Facebook interactions were more common on fake news sites than on reference news media. Bakir and McStay [33] directly blamed Facebook for the appearance of the phenomenon, with the claim that "its seeds were laid in 2010 when Facebook introduced its newsfeed algorithm" (p. 155). This explains the great interest among academics in analyzing Facebook to study fake news and disinformation. However, some studies have already analyzed Twitter's role. For example, Allcott, Gentzkow, and Yu [32] observed that, although still far from the levels of interaction on Facebook, interactions with fake news sites on Twitter have been notably growing. Similarly, and despite the limitations of academic research due to privacy, WhatsApp has been considered a problematic platform for the diffusion of fake news content [47]. In light of these works, we pose the following research question:

**Research Question 1a (RQ1a):** In what social media do Spanish citizens perceive the greatest presence of fake news?

To obtain a more complete perspective, and due to the possible existence of differences between perceptions and real experience, a secondary question is posed:

**Research Question 1b (RQ1b):** In what social media have Spanish citizens experienced the greatest presence of fake news?

Finally, to serve as a basis for designing strategies to fight fake news and disinformation, it is necessary to more deeply analyze the possible factors underlying the different experiences related to this phenomenon, such as the use of social media, ideology, socioeconomic or educational level, age, and gender. The most relevant works in this area were conducted in the U.S. and dealt more with the propensity to believe or share fake news [43,48] than the perceptions of different media types. The closest work to ours was published by Masip, Suau, and Ruiz-Caballero [11], who studied the effects of ideology on trust in different media in Spain. Following this line of research, but with a broader approach, we pose the following question:

**Research Question 2a (RQ2a):** Do gender, age, educational level, socioeconomic level, or ideology affect the perceived or experienced presence of fake news?

Among the factors that might have an influence, the use of social media should be highlighted, as past studies [49] observed that a more frequent use of Facebook correlates with the greatest consumption of fake news. Thus, the following research question is presented:

**Research Question 2b (RQ2b):** How does the frequency of social media use affect the perceived and experienced presence of fake news?

## 3. Materials and Methods

This work follows the design of a study by Blanco-Herrero and Arcila-Calderón [9] on the perceptions of Spanish journalists but instead places the focus on the general Spanish adult population. Together with this questionnaire, the questionnaires used in the Worlds of Journalism Study and the Encuesta de Percepción Social de la Ciencia of the Spanish Foundation for Science and Technology [50] were used as models for the design of the questions, alongside the suggestions of experts during the validation process. The survey was conducted in October 2020, using the Qualtrics platform for design and distribution. The questionnaire was validated in two phases. First, validation was performed by a group of experts in the field—mostly members of Maldita Migración, belonging to the Maldita.es foundation, one of the most well-known fact-checking groups in Spain. Second, a pilot test was used to measure the reliability and stability of the instrument. For the pilot test, a subsample of 32 people answered the questionnaire twice, with 10–15 days between each response. This process allowed the removal or modification of items that offered lower Intraclass Coefficient (ICC) values.

Once validated, distribution of the questionnaire took place between the 7th and 12th of October using a subcontracted panel from Qualtrics to ensure the quality and adequacy of the sample. The total number of responses was 423, but two responses were removed as they did not meet the prerequisites of being Spanish and/or an adult. The final sample included 421 people, with 50.1% women and 49.4% men. The mean age was 34.27 years (SD = 12.577). The ideologies of the respondents, although balanced, tended to lean slightly towards the left (M = 4.55; SD = 2.512, with values from 1 (extreme left) to 10 (extreme right)). The level of education was measured with seven categories, with the most common being a university, masters, or postgraduate education (40.6% of the sample). Finally, family income was measured with five levels, with the most common group taking home after-tax family income over (26.3%) or around (33.5%) 1100 EUR per month.

### 3.1. Measures

The questionnaire used for this study was part of a broader survey that included, in addition to fake news, questions about hate speech and other factors related to the phenomenon. The questions chosen to assess our hypothesis and answer our research questions were grouped into three categories. The first category sought to identify the sociodemographic features of each respondent with questions on gender, age, level of education (no studies; primary school or equivalent; secondary school or equivalent; *Bachillerato* or equivalent; vocational training or equivalent; university degrees, masters, or postgraduate studies; and third-cycle studies (doctorate)), family income (assuming around 1100 EUR as family median income after taxes, the possible answers were very inferior, less than half; inferior; around that amount; superior; very superior, or more than two times higher), and political ideology (between 1 (extreme left)) and 10 (extreme right)). This section also included three questions to determine what type(s) of social media the person uses, as well as the frequency of his or her use of social and other types of media as sources of information. In each case, social media use was measured between 1 (never) and 5 (several times a day).

The second part of the questionnaire included two questions with several items to determine citizens' perception of fake news in different types of media (social media, digital media, blogs, press, radio, television, and interpersonal communication) and different social media platforms (Facebook, Twitter, Instagram, YouTube, LinkedIn, Telegram, TikTok, and WhatsApp). The perceived presence of fake news was measured from 1 (no fake news) to 5 (a great deal of fake news). Finally, the last section measured each respondent's personal experience with fake news on social media by asking whether the surveyed individual had encountered any content on social media that he or she believed to be fake; only previously selected social media platforms appeared as a choice. The reception of this content was measured between 1 (never) to 5 (many times).

### 3.2. Analysis

The 421 valid answers were anonymized and analyzed using version 26 of IBM's SPSS. First, to confirm that no inconsistencies were present, an exploratory analysis of the data was conducted, and the graphical distribution of frequencies was checked. The central part of the analysis included comparisons of the means using Student's T test for independent samples, one-way ANOVA and repeated-measures ANOVA tests, and Pearson's correlations. Type I errors were determined at 95% ($\alpha = 0.05$).

## 4. Results

The most commonly used social platform is WhatsApp, used by 87.6% (n = 369) of the sample. It is followed by YouTube (77.7%, n = 327), Facebook (75.1%, n = 326), and Instagram (75.1%, n = 316). These four are also the ones with a greater frequency of use: WhatsApp (M = 4.81; SD = 0.606), followed by Instagram (M = 4.49; SD = 0.903), YouTube (M = 4.11; SD = 1.003), and Facebook (M = 4.01; SD = 1.140). Twitter offered intermediate values, being used by 49.2% of the sample (n = 207) and with a frequency of use similar to Facebook (M = 4.01; SD = 1.140). Far from them, both in terms of users and frequency of use, we can find TikTok, Telegram, and LinkedIn, as Table 1 shows in detail.

Regarding the media used as a source of information, it can be observed that interpersonal communication (including instant messaging services, such as WhatsApp) is the most used (M = 3.94; SD = 1.301), followed by social media (M = 3.67; SD = 1.387) and television (M = 3.66; SD = 1.218). In an intermediate place we find digital media (M = 2.97; SD = 1.377), and with less frequency of use we can find printed media (M = 2.60; SD = 1.284), radio (M = 2.58; SD = 1.331), and blogs (M = 2.16; SD = 1.201). This is relevant because it shows the declining relevance of traditional media in the current information scenario [33], in which, in the context of post-truth [51], citizens trust alternative channels to find information.

**Table 1.** Most used social media.

| Social Media | Users % | Frequency of Use (M, SD) |
|---|---|---|
| Facebook | **77.4% (n = 326)** | 4.01 (1.061) |
| Twitter | 49.2% (n = 207) | 4.01 (1.140) |
| Instagram | 75.1% (n = 316) | **4.49 (0.903)** |
| YouTube | **77.7% (n = 327)** | **4.11 (1.003)** |
| LinkedIn | 25.2% (n = 106) | 3.31 (1.041) |
| Telegram | 23.3% (n = 98) | 3.98 (1.140) |
| TikTok | 33.5% (n = 141) | 3.85 (1.125) |
| WhatsApp | **87.6% (n = 369)** | **4.81 (0.606)** |

Source: the authors.

### 4.1. Perceived and Experienced Presence of Fake News

In response to H1, we determined that the media types with the greatest perceived presence of fake news are interpersonal communication (M = 3.63; SD = 1.126) and social media (M = 3.73; SD = 1.031). On the other side, the perception was lowest among radio (M = 2.72; SD = 0.987) and printed media (M = 2.90; SD = 1.093). In between these results are digital media (M = 3.01; SD = 1.013), television (M = 3.28; SD = 1.165), and blogs (M = 3.34; SD = 1.074). These differences are significant [$F_{(6)} = 83.765$, $p < 0.001$, $h^2_p = 0.166$] and show that the greatest presence of fake news is found in the most commonly used media.

In response to RQ1a, we observed a similar phenomenon for social media, as the greatest presence of fake news was perceived on WhatsApp (M = 3.82; SD = 1.034) and Facebook (M = 3.81; SD = 1.060). Fake news has a notable presence on Twitter (M = 3.71; SD = 1.043) and Instagram (M = 3.71; SD = 1.004), followed by TikTok (M = 3.57; SD = 1.097), YouTube (M = 3.52; SD = 1.011), and Telegram (M = 3.38; SD = 1.050), with LinkedIn in last place (M = 3.11; SD = 1.086), as the social network with the smallest presence of fake news. These differences were also significant [$F_{(7)} = 41.902$, $p < 0.001$, $h^2_p = 0.091$].

In response to RQ1b, smaller values were observed when respondents were asked about their own experiences, showing that personally experienced fake news was less common than perceptions of such news. Despite this decrease, the order of the results did not change, except for one case. WhatsApp (M = 3.57; SD = 1.164) and Facebook (M = 3.56; SD = 1.013) again featured the greatest presence of fake news, but these platforms were overtaken by Twitter (M = 3.61; SD = 1.013) as the social network where the most fake news was experienced by the sample. Instagram (M = 3.25; SD = 1.105), YouTube (M = 3.03; SD = 1.116), TikTok (M = 2.93; SD = 1.313), and Telegram (M = 2.87; SD = 1.224) again assumed intermediate positions, with LinkedIn as the social media platform with the smallest presence of fake news (M = 2.29; SD = 1.207). These differences were also significant [$F_{(7)} = 2.966$, $p < 0.01$, $h^2_p = 0.175$]. Moreover, even though the level of significance was smaller (likely because of the few cases on some of the least commonly used social platforms), the effect size was the largest, showing that the strongest differences can be found in experience, rather than perception.

### 4.2. Differences Based on Personal Features

To answer RQ2a, we must explore the potential differences based on gender. First, the women (M = 30.17; SD = 10.076) in the sample were significantly younger than the men (M = 38.64; SD = 13.484), [$t_{(381.280}} = 7.266$, $p < 0.001$, d = 0.71], and the family incomes reported by the women (M = 2.72; SD = 0.912) were significantly smaller than those of the men (M = 2.45; SD = 0.972), [$t_{(398)} = -2.909$, $p < 0.01$, d = 0.29]. At the same time, women (M = 4.35; SD = 2.645) tended to be located further on the political left than men (M = 4.76; SD = 2.355), [$t_{(396.696)} = 1.649$, $p = 0.1$].

The use of Facebook also tended to be higher among women (M = 4.12; SD = 1.042) than among men (M = 3.90; SD = 1.074), [$t_{(322)} = -1.883$, $p = 0.061$]. Further, women (M = 4.69; SD = 0.739) used Instagram significantly more often than men (M = 4.20; SD = 1.026), [$t_{(222.833)} = -4.701$, $p < 0.001$, d = 0.55]. The same result was observed for WhatsApp, with a significantly higher use among women (M = 4.89; SD = 0.506) than among men (M = 4.73;

SD = 0.693), [t(309.713) = −2.492, $p < 0.05$, d = 0.26]. However, men (M = 4.18; SD = 1.108) tended to use Telegram more frequently than women (M = 3.76; SD = 1.158), [t(95) = 1.805, $p = 0.074$]. In general terms, the use of social media as a source of information was found to be more common among women (M = 4.00; SD = 1.221) than among men (M = 3.36; SD = 1.461), [t(402.232) = −4.859, $p < 0.001$, d = 0.48]. Women (M = 4.21; SD = 1.153) also used interpersonal communication more frequently than men (M = 3.68; SD = 1.386) as a source of information [t(401.657) = −4.295, $p < 0.001$, d = 0.42]. On the other hand, men (M = 2.82; SD = 1.276) consumed printed media more often than women (M = 2.37; SD = 1.248) as a source of information [t(417) = 3.630, $p < 0.001$, d = 0.36]. The same result was observed for radio, which was more commonly used by men (M = 2.87; SD = 1.307) than by women (M = 2.30; SD = 1.295), [t(417) = 4.497, $p < 0.001$, d = 0.44].

The presence of fake news on social media was perceived as significantly higher by women (M = 3.85; SD = 0.918) than by men (M = 3.63; SD = 1.113), [t(400.337) = −2.190, $p < 0.05$, d = 0.22]. Women (M = 3.23; SD = 1.125) also perceived a greater presence of fake news in television than men (M = 2.98; SD = 1.216), [t(417) = −2.198, $p < 0.05$, d = 0.21]. The same result was observed for interpersonal communication, through which women (M = 3.74; SD = 1.110) tended to find more fake content than men (M = 3.52; SD = 1.133), [t(417) = −1.965, $p = 0.05$]. More specifically, women (M = 3.93; SD = 0.966) perceived more fake news on Facebook than men (M = 3.70, DT = 1.141), [t(404.018) = −2.196, $p < 0.05$, d = 0.22].

Regarding personal experience, women (M = 3.65; SD = 1.018) claimed to have encountered fake news on Facebook more often than men (M = 3.45; SD = 1.073), [t(324) = −1.754, $p = 0.080$]. At the same time, women (M = 3.68; SD = 1.159) tended to have encountered more fake news on WhatsApp than men (M = 3.45; SD = 1.161), [t(363) = −1.943, $p = 0.053$], whereas men (M = 3.18; SD = 1.178) encountered significantly more fake content on Instagram than women (M = 2.50; SD = 1.188), [t(95) = 2.813, $p < 0.01$, d = 0.57].

Besides the aforementioned younger ages of the women in the sample compared to the men, age was found to significantly correlate with family income [R(401) = 0.167, $p < 0.05$], which was higher among older people. A significant and negative correlation was found between the frequency of using social media as an information source and age; that is, social media is more commonly used among younger people [R(420) = −0.244, $p < 0.001$]. Similarly, the frequency of Instagram use significantly increased as the age of the surveyed person decreased [R(312) = −0.319, $p < 0.001$]. Moreover, the frequency of using blogs as a source of information significantly correlated in a negative way with age [R(420) = −0.149, $p < 0.01$]. Surprisingly, no correlation was found between age and the use of printed media—something that was observed with the consumption of radio, which was positively correlated with age—that is, older people listened to radio more often to find information [R(420) = 0.179, $p < 0.001$]. Finally, the frequency of using interpersonal communication as a source of information correlated significantly and negatively with age, as younger people were found to use this channel more often than older age groups [R(420) = −0.103, p < 0.05].

A negative correlation was observed between age and the perception of fake news on television, indicating a greater skepticism towards this medium among younger people [R(420) = −0.176, $p < 0.001$]. On the other hand, the presence of fake news on Facebook was perceived to be higher among older people [R(420) = 0.104, $p < 0.05$]. The same result was observed for LinkedIn [R(420) = 0.132, $p < 0.01$] and Telegram [R(420) = 0.124, $p < 0.05$]. The correlation between the experience of having encountered fake content on social media and age was significant and negative for Facebook [R(325) = −0.177, $p < 0.01$], Instagram [R(314) = −0.193, $p < 0.01$], YouTube [R(324) = −0.165, $p < 0.01$], LinkedIn [R(105) = −0.237, $p < 0.05$], TikTok [R(141) = −0.167, $p < 0.05$], and WhatsApp [R(364) = −0.214, $p < 0.001$]; the same result was observed, as a trend, for Telegram [R(97) = −0.171, $p = 0.095$].

Level of education was found to correlate significantly with family income [R(399) = 0.229, $p < 0.001$] and political ideology, [R(401) = 0.102, $p < 0.05$]; i.e., people with higher education levels were observed to have higher incomes and a tendency to be located more on the right

side of the political spectrum. The level of education also correlates in a significant and positive way with the frequency of using social media [R(416) = 0.103, *p* < 0.05], digital media [R(416) = 0.179, *p* < 0.001], blogs [R(416) = 0.139, *p* < 0.01], and radio [R(416) = 0.101, *p* < 0.05] as sources of information.

A significant and positive correlation was also observed between the level of education and the perceived presence of fake news on social media [R(416) = 0.165, *p* < 0.01], blogs [R(416) = 0.183, *p* < 0.001], and interpersonal communication [R(416) = 0.125, *p* < 0.05], as well as on Facebook [R(416) = 0.123, *p* < 0.05] and Twitter [R(416) = 0.133, *p* < 0.01]. At the same time, on Twitter, a trend and positive correlation was observed only between the level of education and the experienced presence of fake news [R(204) = 0.125, *p* = 0.075].

Besides significant correlations between level of education and age, as well as the lower family income among women, this economic variable showed a trend and positive correlation with the frequency of digital media use [R(402) = 0.103, *p* = 0.093]. This correlation was significant for radio [R(402) = 0.129, *p* < 0.05] and television [R(402) = 0.110, *p* < 0.05].

Family income tends to negatively correlate with the perceived presence of fake news in digital media [R(402) = −0.085, *p* = 0.087], and this correlation was found to be significant for the perceived presence of fake news on the radio [R(402) = −0.151, *p* < 0.01]. Focusing on social media, this correlation was found to be significant and positive for YouTube [R(402) = 0.114, *p* < 0.05] and WhatsApp [R(402) = 0.117, *p* < 0.05] but was a trend only for Telegram [R(402) = 0.090, *p* = 0.073]. Similarly, a significant and negative correlation was observed between family income and the experience of having encountered fake news on Instagram [R(303) = −0.114, *p* < 0.05] and LinkedIn [R(103) = −0.198, *p* < 0.05].

The last analyzed factor was the influence of political ideology. A significant and negative correlation was found between ideology and the frequency of Twitter use, which means that this social platform is more commonly used by citizens on the ideological left [R(199) = −0.211, *p* < 0.01]. This phenomenon was also observed for Instagram, where the correlation was only a trend [R(305) = −0.095, *p* = 0.098]. On the other hand, the use of blogs [R(406) = 0.101, *p* < 0.05], printed media [R(406) = 0.190, *p* < 0.001], radio [R(406) = 0.107, *p* < 0.05], television [R(406) = 0.148, *p* < 0.05], and interpersonal communication [R(406) = 0.109, *p* < 0.05] as sources of information was positively and significantly correlated with ideology; these media were found to be more commonly consumed by those on the right of the political–ideological spectrum.

The correlation between political ideology and the perceived presence of fake news on interpersonal communication was significantly negative [R(406) = −0.103, *p* < 0.05]. The same result was observed for Facebook [R(406) = −0.110, *p* < 0.05], which indicates that more progressive individuals believe that there is more fake news on these media platforms.

In response to RQ2b, for all types of media, positive correlations seem to exist between the perceived presence of fake news and the frequency of use, although these correlations were only found to be significant for interpersonal communication, social media, and printed media, always with small effect sizes. A similar result was observed for the various social media, among which significant correlations were found for Twitter, Instagram, WhatsApp, and Facebook, where the effect sizes were only slightly higher. In the case of social media, the experienced presence of fake news was also positively correlated with the frequency of use, and these correlations were found to be significant for TikTok, Facebook, YouTube, and WhatsApp; the effect sizes were, moreover, similar to those observed between frequency of use and the perceived presence of fake news. All these values can be found in Table 2.

Finally, positive correlations were observed between the perceived and experienced presence of fake news on social media. This correlation was found to be significant for Facebook [R(326) = 0.345, *p* < 0.001], Twitter [R(204) = 0.323, *p* < 0.001], Instagram [R(315) = 0.265, *p* < 0.001], YouTube [R(325) = 0.308, *p* < 0.001], LinkedIn [R(105) = 0.319, *p* < 0.001], and WhatsApp [R(365) = 0.364, *p* < 0.001]. For Telegram [R(98) = 0.233, *p* < 0.05] and TikTok [R(141) = 0.311, *p* < 0.05], the same result was observed, but the level of

significance decreased below 0.05 due to the smaller number of cases, as these social media platforms have fewer users.

**Table 2.** Correlations between frequency of use and perceived and experienced presence of fake news in different media.

|  | Perceived Presence | Experienced Presence |
|---|---|---|
| Social media | 0.146 ** | - |
| Digital media | 0.135 | - |
| Blogs | 0.039 | - |
| Printed media | 0.131 ** | - |
| Radio | 0.050 | - |
| Television | −0.020 | - |
| Interpersonal communication | 0.179 ** | - |
| Facebook | 0.126 * | 0.144 ** |
| Twitter | 0.240 ** | 0.132 |
| Instagram | 0.165 ** | 0.068 |
| YouTube | 0.107 | 0.143 * |
| LinkedIn | 0.024 | 0.104 |
| Telegram | 0.130 | 0.065 |
| TikTok | 0.112 | 0.221 ** |
| WhatsApp | 0.151 ** | 0.112 * |

Source: the authors. * $p > 0.05$; ** $p > 0.01$.

## 5. Discussion

The study partially confirmed H1 because even though the greatest presence of fake news was perceived in interpersonal communication, most fake news was perceived on social media platforms. This result agrees with previous research [9,19,43] confirming the connection between the problem of fake news and social media. It should be noted that interpersonal communication, which is conducted between peers using a private channel, often takes place on platforms such as WhatsApp and Telegram.

WhatsApp was, in fact, one of the social media platforms on which the strongest presence of fake news was perceived—a result that matches previous observations [47,52,53] and confirms the concerns of previous studies in a Spanish setting [54]. Further answering RQ1a and RQ1b, Facebook and Twitter, together with WhatsApp, were found to be the platforms with the greatest perception of fake news, while Twitter provided the most experienced fake news. This result also matches previous observations, such as those of Allcott, Gentzkow, and Yu [32] noting that while on Facebook "the overall magnitude of the misinformation problem may have declined, possibly due to changes to the Facebook platform following the 2016 election", the increase of the problem on Twitter remains relevant.

Given the existing limitations in accurately measuring the presence of fake news, this study offers an alternative way to evaluate which platforms and types of media feature the greatest presence of fake news. Efforts to counter fake news and disinformation should, therefore, focus on these platforms. Fighting fake news is easier on Facebook and Twitter, which allow the removal of profiles or content, the flagging of disputed content, and links to fact checks. However, this process is more difficult on WhatsApp due to its privacy and infrastructure, although limitations to message forwarding and the introduction of notifications when a message has been forwarded multiple times have attempted to slow the spread of rumors and fake news.

Age and gender were found to be the most relevant factors that affected the perceptions of citizens, with few specific differences based on education level, family income, or political ideology. These differences based on personal features could be related to each other. For example, women tended to be younger and more left-wing, which are characteristics correlated to a more negative perception regarding the presence of fake news. At the same time, the experienced presence of fake news was found to be affected only by age and gender, with even smaller effects observed for the other factors.

The greater level of skepticism observed among young people and women—and, to a lesser extent, among more highly educated and left-wing people—corresponded with the observed factors influencing the credibility and sharing probability of fake news [43,48,55]. To some extent, this correlation could explain the reasons for these previous observations: If a person's perception of a platform is more negative and that person is aware of the potential presence of fake content, he or she is less prepared to believe or share that content [56]. Although further work is needed in this area, our study contributes to the existing literature on the credibility and spread of fake news.

Although not part of the study, some of the strongest differences were previously found not in the perceived or experienced presence of fake news but in the frequency of using different types of media and social media. For example, political ideology strongly influences what types of media or social media are consumed but does not strongly influence the perceptions or experiences related to the presence of fake news. These observations are not surprising: Young people use Instagram and social media more frequently, and people with higher levels of education and/or income consume more media in general. However, these observations also interact with other variables and help to explain people's perceptions and experiences with fake content. For example, when observing the relevance of age, experience was found to be higher among young citizens, but the perceptions were more negative among older individuals, showing that the greater media and digital literacy among young people might help them better identify fake content.

Another relevant aspect could be the influence of frequency of use (which was higher among younger people) in the experienced presence of fake content. Thus, in response to RQ2b, we argue that the use of media positively correlates with the perceived and experienced presence of fake news. The strongest presence of fake news was found among the most commonly used channels (interpersonal communication and social media) and social media platforms (Facebook and WhatsApp), while the experienced presence was highest on Twitter, followed by Facebook and WhatsApp, which are all frequently used platforms. This was also observed in previous works [49]. Moreover, it makes sense that users perceive more fake content in the media they most frequently consume. At the same time, this result indicates that the presence of fake news does not lead to a decrease in the usage or abandonment of a type of media or social platform, something that would be expected were that media not seen as trustworthy or reliable. The decision to ask all respondents (not only the users of the media type or social platform in question) about the perceived or imagined presence of fake news was done to measure whether some types of media or social networks have a generalized negative image that could keep people from using them. This result was not observed. In turn, for this RQ, the use and frequency of use of media seemed to most strongly determine the perceived and experienced presence of fake content. Further studies will be needed to further analyze the causes and implications of this observation.

In general, all the effect sizes of the correlations found in RQ2 were small. Thus, these observations could help design more adequate strategies against fake news, but further analysis is needed. An additional limitation of this work is that the general frequency of media use was not analyzed, as this question was only presented in connection to different types of social media. We chose not to explore this factor in order to ensure the brevity of the questionnaire and to highlight the different types of social media, which are key platforms for the spread of fake news, as the literature and the present study demonstrate.

Lastly, this study did not seek to measure the amount of fake news on social media. Previous studies quantifying online disinformation observed that fake news and disinformation, despite their relevance and potential harm, are only a small part of the conversation on social media and are often connected to partisan media [27]. However, the need to fight this disruptive and dangerous phenomenon is unquestionable. The present study sought to provide more detailed knowledge of the approaches taken by Spanish citizens towards

fake news, with the ultimate goal of helping to design strategies that could reduce the spread of, and belief in, this content.

The observations of this survey are partially limited by the time at which it was conducted (October 2020), during the Covid-19 pandemic, a situation in which the attention paid to disinformation has been strong and might have influenced the answers; for this reason, future works are needed to help study the longitudinal evolution of these observations.

As a conclusion, social media and interpersonal communication seem to be the scenarios for the largest spread of fake news, partially confirming H1; Twitter, WhatsApp, and Facebook seem to be the platforms with a greater presence of misinformation (RQ1); and women, younger people, and, although less strongly, more educated and progressive people seem to perceive a greater presence of fake news; this perception and experience seems to also be higher for people who use more of the media in question.

**Author Contributions:** Conceptualization, D.B.-H., J.J.A. and P.S.-H.; methodology, D.B.-H., J.J.A. and P.S.-H.; software, D.B.-H., J.J.A. and P.S.-H.; validation, D.B.-H., J.J.A. and P.S.-H.; formal analysis, D.B.-H., J.J.A. and P.S.-H.; investigation, D.B.-H., J.J.A. and P.S.-H.; resources, D.B.-H., J.J.A. and P.S.-H.; data curation, D.B.-H., J.J.A. and P.S.-H.; writing—original draft preparation, D.B.-H., J.J.A. and P.S.-H.; writing—review and editing, D.B.-H., J.J.A. and P.S.-H.; visualization, D.B.-H., J.J.A. and P.S.-H.; supervision, D.B.-H., J.J.A. and P.S.-H.; project administration, D.B.-H., J.J.A. and P.S.-H.; funding acquisition, D.B.-H., J.J.A. and P.S.-H. All authors have read and agreed to the published version of the manuscript.

**Funding:** This research was funded by the Spanish Foundation for Science and Technology (FECYT) through the Call for grants for the promotion of scientific, technological, and innovation culture 2018–2019 (reference FCT-18-13437) and by the FPU18/01455 Grant of the Ministry of Universities of Spain.

**Institutional Review Board Statement:** Not applicable.

**Informed Consent Statement:** Not applicable.

**Data Availability Statement:** The data presented in this study are available within the article.

**Acknowledgments:** The authors thank Carlos Arcila-Calderón, PI of the project that produced this research, for his guidance.

**Conflicts of Interest:** The authors declare no conflict of interest.

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
