# Peer review of "Citizen Perceptions of Fake News in Spain: Socioeconomic, Demographic, and Ideological Differences"

_publications, doi:10.3390/publications9030035_

Round 1
Reviewer 1 Report
No comments.
Author Response
We would like to thank his/her time and review.
Reviewer 2 Report
The research topic is interesting. Moreover, the paper in general terms is well written, and has a pertinent topic.
A general concern is about to the results and its discussion with the literature. It should be more detailed and more well fundamented.
The authors should consider the following recommendations in order to improve the original manuscript:
- To include the structure of the paper in the Introduction section.
- Introduction and Literature review should be two separate sections, not a common mixed and incoherent section as now. To include the structure of the paper in the new Introduction section. It is more than necessary to include a new section "Literature review". The authors also did not provide sufficient evidence on literature review to support the hypotheses. The Introduction section also includes the Literature review section which is practically non-existent being mentioned only a few bibliographic references quite uncorrelated. Authors should take into consideration much more recent publications in the sphere of discussed subject matter, especially studies conducted during the last 5 years.
- Deepen the description of the limitations of conducted research and indicate the trends for further empirical research.
- To expand the managerial implications in the article.
- The sources must be added under each table and figure.
- The conclusions section needs to be greatly improved and expanded.
- Human proofreading, English grammar and spelling correction are also required in order to improve the quality of the manuscript.
- I would also like to see a well-developed discussion comparing and contrasting solution/results presented in the work with existing work and then a subsection of it presenting contributions to theory/knowledge/literature and followed by a subsection on “Implications for practice”.
Regarding consumer behaviour framework but also on socioeconomic, demographic and ideological differences new perspectives, includind the impact of COVID-19 pandemic, I suggest extending the literature section by including at least the following relevant studies:
- Hawaldar, I.T.; Ullal, M.S.; Birau, F.R.; Spulbar, C.M. Trapping Fake Discounts as Drivers of Real Revenues and Their Impact on Consumer’s Behavior in India: A Case Study. Sustainability 2019, 11, 4637.
- Bonera, M.; Corvi, E.; Codini, A.P.; Ma, R. Does Nationality Matter in Eco-Behaviour? Sustainability 2017, 9, 1694.
- Batool, M., Ghulam, H., Hayat, M.A., Naeem, M.Z., Ejaz, A., Imran, Z.A., Spulbar, C., Birau, R. & Gorun, T.H. (2020) How COVID-19 has shaken the sharing economy? An analysis using Google trends data, Economic Research-Ekonomska Istraživanja, https://www.tandfonline.com/doi/full/10.1080/1331677X.2020.1863830.
- Antonides, G. Sustainable Consumer Behaviour: A Collection of Empirical Studies. Sustainability 2017, 9, 1686.
- Ullal, M.S., Spulbar, C., Hawaldar, I.T., Popescu, V. & Birau, R. (2021) The impact of online reviews on e-commerce sales in India: a case study, Economic Research-Ekonomska Istraživanja, DOI: 10.1080/1331677X.2020.1865179, https://www.tandfonline.com/doi/full/10.1080/1331677X.2020.1865179.
- Qaiser Gillani, D.; Gillani, S.A.S.; Naeem, M.Z.; Spulbar, C.; Coker-Farrell, E.; Ejaz, A.; Birau, R. (2021) The Nexus between Sustainable Economic Development and Government Health Expenditure in Asian Countries Based on Ecological Footprint Consumption. Sustainability, 13, 6824. https://doi.org/10.3390/su13126824.
- Hayat, M.A.; Ghulam, H.; Batool, M.; Naeem, M.Z.; Ejaz, A.; Spulbar, C.; Birau, R. (2021) Investigating the Causal Linkages among Inflation, Interest Rate, and Economic Growth in Pakistan under the Influence of COVID-19 Pandemic: A Wavelet Transformation Approach, Journal of Risk and Financial Management, 14(6):277. https://doi.org/10.3390/jrfm14060277.
Author Response
We thank and appreciate the comments of the reviewer. Next we address the suggested changes.
To include the structure of the paper in the Introduction section.
A new paragraph has been added at the end of the introduction section (lines 57-61): “This article addresses and narrows the discussion about fake news, especially in a Spanish setting. First, we detail the process followed to study the perceptions of Spanish citizens on this issue. Then, we explore the obtained results and, finally, discuss the results in connection to the existing literature in the field while also describing the study’s limitations and future lines of work.”
Introduction and Literature review should be two separate sections, not a common mixed and incoherent section as now. It is more than necessary to include a new section "Literature review". The authors also did not provide sufficient evidence on literature review to support the hypotheses. The Introduction section also includes the Literature review section which is practically non-existent being mentioned only a few bibliographic references quite uncorrelated. Authors should take into consideration much more recent publications in the sphere of discussed subject matter, especially studies conducted during the last 5 years.
Both sections have been separated and a wider review of the literature has been conducted, adding around 20 new references.
Deepen the description of the limitations of conducted research and indicate the trends for further empirical research.
Although several limitations and future lines of work were present along the discussion section, an additional reference has been made in lines 504-508: “The observations of this survey are partially limited by the moment in which it was conducted –in October 2020–, during the Covid-19 pandemic, a situation in which the attention paid to disinformation has been strong and might have influenced the answers; for this reason, future works are needed to help studying the longitudinal evolution of these observations.”
To expand the managerial implications in the article.
The authors believe that the practical implications of the study are clear and addressing managerial issues would make the article lose its focus.
The sources must be added under each table and figure.
All tables have a source.
The conclusions section needs to be greatly improved and expanded.
This section has been expanded and improved: more discussion has been added mostly in lines 472-487 (“The strongest presence of fake news was found among the most commonly used channels (interpersonal communication and social media) and social media platforms (Facebook and WhatsApp), while the experienced presence was highest on Twitter, followed by Facebook and WhatsApp, which are all frequently used platforms. This was also observed in previous works [50]. Moreover, it makes sense that users perceive more fake content in the media they most frequently consume. At the same time, this result indicates that the presence of fake news does not lead to a decrease in the usage or abandonment of a type of media or social platform, something that would be expected were that media not seen as trustworthy or reliable. The decision to ask all respondents (not only the users of the media type or social platform in question) about the perceived or imagined presence of fake news was done to measure whether some types of media or social networks have a generalized negative image that could keep people from using them. This result was not observed. In turn, for this RQ, the use and frequency of use of media seemed to most strongly determine the perceived and experienced presence of fake content. Further studies will be needed to further analyze the causes and implications of this observation.”) and in lines 496-514 (“Lastly, this study does not seek to measure the amount of fake news on social media. Previous studies quantifying online disinformation observed that fake news and disin-formation, despite their relevance and potential harm, are only a small part of the con-versation on social media and are often connected to partisan media [27]. However, the need to fight this disruptive and dangerous phenomenon is unquestionable. The present study sought to provide more detailed knowledge about the approaches taken by Spanish citizens towards fake news, with the ultimate goal of helping to design strategies that could reduce the spread of, and belief in, this content. The observations of this survey are partially limited by the moment in which it was conducted –in October 2020–, during the Covid-19 pandemic, a situation in which the attention paid to disinformation has been strong and might have influenced the answers; for this reason, future works are needed to help studying the longitudinal evolution of these observations. As a conclusion, social media and interpersonal communication seem to be the scenarios for the largest spread of fake news, partially confirming H1; Twitter, WhatsApp and Facebook seem to be the platforms with a greater presence of misinformation (RQ1); and women, younger people and, although less strongly, more educated and progressist people seem to perceive a greater presence of fake news; this perception and experience seems to be also higher by people who use more the media in question.”)
Human proofreading, English grammar and spelling correction are also required in order to improve the quality of the manuscript.
A professional proofreading has been conducted by the MDPI editing service on the complete article.
I would also like to see a well-developed discussion comparing and contrasting solution/results presented in the work with existing work and then a subsection of it presenting contributions to theory/knowledge/literature and followed by a subsection on “Implications for practice”.
The discussion has been broadened and a stronger focus on practical implications has been incorporated. This can be seen in lines 496-503: “Lastly, this study does not seek to measure the amount of fake news on social media. Previous studies quantifying online disinformation observed that fake news and disin-formation, despite their relevance and potential harm, are only a small part of the con-versation on social media and are often connected to partisan media [27]. However, the need to fight this disruptive and dangerous phenomenon is unquestionable. The present study sought to provide more detailed knowledge about the approaches taken by Spanish citizens towards fake news, with the ultimate goal of helping to design strategies that could reduce the spread of, and belief in, this content”
Regarding consumer behaviour framework but also on socioeconomic, demographic and ideological differences new perspectives, includind the impact of COVID-19 pandemic, I suggest extending the literature section by including at least the following relevant studies:
- Hawaldar, I.T.; Ullal, M.S.; Birau, F.R.; Spulbar, C.M. Trapping Fake Discounts as Drivers of Real Revenues and Their Impact on Consumer’s Behavior in India: A Case Study. Sustainability 2019, 11, 4637.
- Bonera, M.; Corvi, E.; Codini, A.P.; Ma, R. Does Nationality Matter in Eco-Behaviour? Sustainability 2017, 9, 1694.
- Batool, M., Ghulam, H., Hayat, M.A., Naeem, M.Z., Ejaz, A., Imran, Z.A., Spulbar, C., Birau, R. & Gorun, T.H. (2020) How COVID-19 has shaken the sharing economy? An analysis using Google trends data, Economic Research-Ekonomska Istraživanja, https://www.tandfonline.com/doi/full/10.1080/1331677X.2020.1863830.
- Antonides, G. Sustainable Consumer Behaviour: A Collection of Empirical Studies. Sustainability 2017, 9, 1686.
- Ullal, M.S., Spulbar, C., Hawaldar, I.T., Popescu, V. & Birau, R. (2021) The impact of online reviews on e-commerce sales in India: a case study, Economic Research-Ekonomska Istraživanja, DOI: 10.1080/1331677X.2020.1865179, https://www.tandfonline.com/doi/full/10.1080/1331677X.2020.1865179.
- Qaiser Gillani, D.; Gillani, S.A.S.; Naeem, M.Z.; Spulbar, C.; Coker-Farrell, E.; Ejaz, A.; Birau, R. (2021) The Nexus between Sustainable Economic Development and Government Health Expenditure in Asian Countries Based on Ecological Footprint Consumption. Sustainability, 13, 6824. https://doi.org/10.3390/su13126824.
- Hayat, M.A.; Ghulam, H.; Batool, M.; Naeem, M.Z.; Ejaz, A.; Spulbar, C.; Birau, R. (2021) Investigating the Causal Linkages among Inflation, Interest Rate, and Economic Growth in Pakistan under the Influence of COVID-19 Pandemic: A Wavelet Transformation Approach, Journal of Risk and Financial Management, 14(6):277. https://doi.org/10.3390/jrfm14060277.
The authors believe that these works do not match the object of study, as they deal with consumer behavior rather than with social and communication implications of fake news. In any case, as requested by the reviewer, a larger set of around 20 new references has been incorporated to the study, including references that broaden the object of study with new theories (“Beyond a terminological discussion, it is relevant to highlight the growth of the problem of disinformation in recent years. This problem is not new, and much discussion around this phenomenon has focused on this factor [22, 23]. Indeed, some of the expla-nations behind the proliferation of fake news utilize traditional theories, such as con-firmation bias [24, 25] and selective exposure [26], as well as traditional communication and media theories, such as agenda-setting [27]. However, other more novel elements and theories should also be considered, including the roles of bots [28] and, very importantly, the roles of echo chambers and filter bubbles [19, 29]. These factors are strongly related to confirmation bias and selective exposure and have been widely discussed in relation to the social media environment [30].”) and references to Covid-19 (“Additionally, many recent works focused on the COVID-19 pandemic [16-18], following an international trend. Building upon these works, the present study seeks to increase knowledge about fake news in a Spanish context”).
Reviewer 3 Report
Review of ‘Citizen perceptions about fake news in Spain: Socioeconomic, demographic and ideological differences’
- The paper intends to ‘identify in which media the perception and experience of the presence of fake news is greatest’, adopting the point of view of consumers and assessing several profile differences.
- The title is fine.
- The abstract and keywords are correct, although the abstract could specify something about the survey target population, maybe addressing the Spanish setting (Spanish adult population?). Also, the number of validated survey answers should be corrected to 421.
- Introduction is structurally correct with the presence of knowledge gaps and research questions. I would only suggest that a paragraph about the Spanish literature on the subject could be brought up. I felt that the Spanish previous studies were scattered.
- I have found no traces of self-plagiarism, despite the paper follows previous work of the authors.
- Line 173 – measures, should be measured.
- Materials and methods section is clear, explaining the questionnaire, although has no mention regarding the origin of the questions.
- In 3.1 (Line 228) it is important to explain that there is a natural correlation between higher use of social media and higher perception of fake news. One cannot perceive if one don’t use it. This should be explored to discuss some of these results. In the Discussion section, this explanation could be inserted after line 433: ‘the use of media positively correlates to the perceived and experienced presence of fake news’.
- The Results and Discussion sections are ok. The discussion opened (in Line 439) is not very relevant, in my opinion. Fake news is a very dangerous threat, but it’s only a small part of the social media ecosystem. The discussion should not be around increase/decrease in the usage, but how to undermine their spreading, like we did with SPAM emails a few years ago.
- Maybe it’s a matter of formatting rules, but I did miss a conclusion of the article, summarizing results and answering all the RQ.
- Congratulations! Well done!
Author Response
We thank and appreciate the comments of the reviewer. Next we address the suggestions.
The abstract and keywords are correct, although the abstract could specify something about the survey target population, maybe addressing the Spanish setting (Spanish adult population?). Also, the number of validated survey answers should be corrected to 421.
Both suggestions have been added to the abstract in lines 16 and 17.
Introduction is structurally correct with the presence of knowledge gaps and research questions. I would only suggest that a paragraph about the Spanish literature on the subject could be brought up. I felt that the Spanish previous studies were scattered.
A new paragraph has been added to the literature review with a revision of the work in the Spanish context. It can be found in lines 48-56: “Besides the interest of the topic of study, this article enrichesexisting quantitative literature by using a deep statistical approach on fake news in a Spanish setting. However, despite not being the most common approach, several works in Spanish academia have applied surveys related to our topic of study, such as the surveys developed by Gualda and Rúas and Masip, Suau, and Ruiz-Caballero [8-11]. Many works have also focused on more specific aspects, including several studies on fact-checking activities [12, 13] and media analyses [14, 15]. Additionally, many recent works focused on the COVID-19 pandemic [16-18], following an international trend. Building upon these works, the present study seeks to increase knowledge about fake news in a Spanish context.”
Line 173 – measures, should be measured.
Changed.
Materials and methods section is clear, explaining the questionnaire, although has no mention regarding the origin of the questions.
The origin of the questions has been referred in lines 184-188: “Together with this questionnaire, the questionnaires used in the Worlds of Journalism Study and the Encuesta de Percepción Social de la Ciencia of the Spanish Foundation for Science and Technology [51] were used as models for the design of the questions, alongside the suggestions of experts during the validation process”
In 3.1 (Line 228) it is important to explain that there is a natural correlation between higher use of social media and higher perception of fake news. One cannot perceive if one don’t use it. This should be explored to discuss some of these results. In the Discussion section, this explanation could be inserted after line 433: ‘the use of media positively correlates to the perceived and experienced presence of fake news’.
The discussion around this aspect has been increased in lines 472-487: “This was also observed in previous works [50]. Moreover, it makes sense that users perceive more fake content in the media they most frequently consume. At the same time, this result indicates that the presence of fake news does not lead to a decrease in the usage or abandonment of a type of media or social platform, something that would be expected were that media not seen as trustworthy or reliable. The decision to ask all respondents (not only the users of the media type or social platform in question) about the perceived or imagined presence of fake news was done to measure whether some types of media or social networks have a generalized negative image that could keep people from using them. This result was not observed. In turn, for this RQ, the use and frequency of use of media seemed to most strongly determine the perceived and experienced presence of fake content. Further studies will be needed to further analyze the causes and implications of this observation.”
The Results and Discussion sections are ok. The discussion opened (in Line 439) is not very relevant, in my opinion. Fake news is a very dangerous threat, but it’s only a small part of the social media ecosystem. The discussion should not be around increase/decrease in the usage, but how to undermine their spreading, like we did with SPAM emails a few years ago.
An additional paragraph was added at the end of section 5. Discussion to clarify this discussion in lines 496-503: “Lastly, this study does not seek to measure the amount of fake news on social media. Previous studies quantifying online disinformation observed that fake news and disin-formation, despite their relevance and potential harm, are only a small part of the con-versation on social media and are often connected to partisan media [27]. However, the need to fight this disruptive and dangerous phenomenon is unquestionable. The present study sought to provide more detailed knowledge about the approaches taken by Spanish citizens towards fake news, with the ultimate goal of helping to design strategies that could reduce the spread of, and belief in, this content”
Maybe it’s a matter of formatting rules, but I did miss a conclusion of the article, summarizing results and answering all the RQ.
A final paragraph has been added at the end of the article (lines 509-514) to include this summary: “As a conclusion, social media and interpersonal communication seem to be the scenarios for the largest spread of fake news, partially confirming H1; Twitter, WhatsApp and Facebook seem to be the platforms with a greater presence of misinformation (RQ1); and women, younger people and, although less strongly, more educated and progressist people seem to perceive a greater presence of fake news; this perception and experience seems to be also higher by people who use more the media in question.”
Reviewer 4 Report
A very interesting and certainly topical paper regarding the phenomenon of misinformation and fake news. A piece of work that I believe merits publication.
I would suggest the following:
Although the literature review is adequate, to achieve a stronger and more complete theoretical section I would suggest seeing in this section a small discussion about
1) the different kinds of fake news/misinformation (video-deepfakes, image content) and in general, the different ways to detect them (e.g. expert-oriented, crowdsourcing-oriented or computational)
2) the different psychological and cognitive theories that can explain the phenomenon and the influence of fake news (e.g. normative influence theory, the confirmation bias, the bandwagon effect)
Finally, I would suggest developing a bit more on the impact of this research in the discussion.
Author Response
We thank and appreciate the comments of the reviewer.
A very interesting and certainly topical paper regarding the phenomenon of misinformation and fake news. A piece of work that I believe merits publication. I would suggest the following: Although the literature review is adequate, to achieve a stronger and more complete theoretical section I would suggest seeing in this section a small discussion about:
1) the different kinds of fake news/misinformation (video-deepfakes, image content) and in general, the different ways to detect them (e.g. expert-oriented, crowdsourcing-oriented or computational)
These elements have been added in a new paragraph to the literature review in lines 108-115: “It should be also noted that, although fake news is usually associated with textual content, such news is also spread through different formats, including images and videos. The emerging technology of deepfakes [35] is one of the most relevant challenges in the current scenario [36]. However, there are already multiple efforts to combat fake news—such as through legislation and media literacy. Among these efforts, the relevance of automatic detection is garnering significant attention due to its already promising results [37]. Alternatives include crowdsourcing detection [38] and the well-established task of fact-checking [39].”
2) the different psychological and cognitive theories that can explain the phenomenon and the influence of fake news (e.g. normative influence theory, the confirmation bias, the bandwagon effect)
This has been included in the literature review, in lines 88-97: “Beyond a terminological discussion, it is relevant to highlight the growth of the problem of disinformation in recent years. This problem is not new, and much discussion around this phenomenon has focused on this factor [22, 23]. Indeed, some of the expla-nations behind the proliferation of fake news utilize traditional theories, such as con-firmation bias [24, 25] and selective exposure [26], as well as traditional communication and media theories, such as agenda-setting [27]. However, other more novel elements and theories should also be considered, including the roles of bots [28] and, very importantly, the roles of echo chambers and filter bubbles [19, 29]. These factors are strongly related to confirmation bias and selective exposure and have been widely discussed in relation to the social media environment [30].”
Finally, I would suggest developing a bit more on the impact of this research in the discussion.
The discussion has been improved with the impact that our research could have. For example, in lines 472-487 (“This was also observed in previous works [50]. Moreover, it makes sense that users perceive more fake content in the media they most frequently consume. At the same time, this result indicates that the presence of fake news does not lead to a decrease in the usage or abandonment of a type of media or social platform, something that would be expected were that media not seen as trustworthy or reliable. The decision to ask all respondents (not only the users of the media type or social platform in question) about the perceived or imagined presence of fake news was done to measure whether some types of media or social networks have a generalized negative image that could keep people from using them. This result was not observed. In turn, for this RQ, the use and frequency of use of media seemed to most strongly determine the perceived and experienced presence of fake content. Further studies will be needed to further analyze the causes and implications of this observation”) or in lines 504-514 (“Lastly, this study does not seek to measure the amount of fake news on social media. Previous studies quantifying online disinformation observed that fake news and disinformation, despite their relevance and potential harm, are only a small part of the con-versation on social media and are often connected to partisan media [27]. However, the need to fight this disruptive and dangerous phenomenon is unquestionable. The present study sought to provide more detailed knowledge about the approaches taken by Spanish citizens towards fake news, with the ultimate goal of helping to design strategies that could reduce the spread of, and belief in, this content. The observations of this survey are partially limited by the moment in which it was conducted –in October 2020–, during the Covid-19 pandemic, a situation in which the attention paid to disinformation has been strong and might have influenced the answers; for this reason, future works are needed to help studying the longitudinal evolution of these observations.”).
Round 2
Reviewer 2 Report
The original manuscript has been significantly improved. The authors followed the recommendations included in the previous review report so that the quality of their research article has greatly increased.